# Innovative Bioactive Ag-SiO_2_/TiO_2_ Coating on a NiTi Shape Memory Alloy: Structure and Mechanism of Its Formation

**DOI:** 10.3390/ma14010099

**Published:** 2020-12-29

**Authors:** Mateusz Dulski, Jacek Balcerzak, Wojciech Simka, Karolina Dudek

**Affiliations:** 1Institute of Materials Engineering, University of Silesia and Silesian Center for Education and Interdisciplinary Research, 75 Pulku Piechoty 1A, 41-500 Chorzow, Poland; 2Department of Molecular Engineering, Faculty of Process and Environmental Engineering, Lodz University of Technology, Wolczanska 213, 90-924 Lodz, Poland; jacek.balcerzak@p.lodz.pl; 3Faculty of Chemistry, Silesian University of Technology, B. Krzywoustego 6, 44-100 Gliwice, Poland; wojciech.simka@polsl.pl; 4Łukasiewicz Research Network-Institute of Ceramics and Building Materials, Refractory Materials Division in Gliwice, Toszecka 99, 44-100 Gliwice, Poland

**Keywords:** silver-silica composite, electrophoretic deposition, NiTi alloy, sintering, mechanism of formation, morphology, structure

## Abstract

In recent years, more and more emphasis has been placed on the development and functionalization of metallic substrates for medical applications to improve their properties and increase their applicability. Today, there are many different types of approaches and materials that are used for this purpose. Our idea was based on a combination of a chemically synthesized Ag-SiO_2_ nanocomposite and the electrophoretic deposition approach on a NiTi shape memory substrate. As a result, silver-silica coating was developed on a previously passivated alloy, which was then subjected to sintering at 700 °C for 2 h. The micrometer-sized coat-forming material was composed of large agglomerates consisting of silica and a thin film of submicron- and nano- spherical-shaped particles built of silver, carbon, and oxygen. Structurally, the coatings consisted of a combination of nanometer-sized silver-carbonate that was embedded in thin amorphous silica and siloxy network. The temperature impact had forced morphological and structural changes such as the consolidation of the coat-forming material, and the partial coalescence of the silver and silica particles. As a result, a new continuous complex ceramic coating was formed and was analyzed in more detail using the XPS, XRD, and Raman methods. According to the structural and chemical analyses, the deposited Ag-SiO_2_ nanocomposite material’s reorganization was due to its reaction with a passivated TiO_2_ layer, which formed an atypical glass-like composite that consisted of SiO_2_-TiO_2_ with silver particles that stabilized the network. Finally, the functionalization of the NiTi surface did not block the shape memory effect.

## 1. Introduction

Metallic materials (e.g., steel, titanium, tantalum, niobium, precious metals) and their alloys have been of interest in medicine, especially in implantology, for over 65 years. This is because these biomaterials are characterized by good mechanical properties and a high degree of biocompatibility [1,2,3,4,5,6]. The NiTi alloys, which have a chemical composition close to equiatomic because of their unique features such as their superelasticity and shape memory effects, are the most popular implant materials used to repair damaged human bones [7,8,9]. The one- and two-way shape memory effects (SME) correspond to the reversible martensitic transformation that is induced by temperature or external stress, which causes a transformation between the B2 parent phase (with cubic structure) and B19′ martensite (with a monoclinic structure). Other properties that enable NiTi alloys’ medical use include their good mechanical properties, corrosion resistance, and biocompatibility [8,9,10,11]. Unfortunately, NiTi alloys also have some limitations that are related to their low thermal stability. More precisely, the high thermal conditions may lead to the decomposition of the B2 parent phase into non-equilibrium (Ni_4_Ti_3_) and equilibrium (Ti_2_Ni and/or Ni_3_Ti) phases [12], as well as affect the shape memory and superelasticity effects [10,13]. Other problems with the NiTi alloy are the possibility of releasing harmful nickel ions into the body environment due to the influence of the long-term effects of human bodily fluids. In turn, the interaction of nickel ions might cause allergic reactions, and as a result, there is a lack of full acceptance of the alloy by the human body [14,15,16].

A solution and an improvement of the applicability of the NiTi alloys may be realized via surface functionalization via the formation of protective coatings that can be prepared in the form of ceramics [17,18,19], polymers [20], metals [21], diamond-like carbon [22], or composites [23,24,25]. An important thing that is worth addressing here is that the materials mentioned above have many advantages and disadvantages. Of all of the materials, the most prospective are ceramics, which have a high biocompatibility level, resist corrosion, and are durable. Polymers also have predictable and repeatable physical and mechanical properties (e.g., Young’s modulus, compressive, and tensile strength) and they bioresorb to non-toxic substances. Metals have favorable mechanical and strength properties, are quite hard, and are resistant to abrasive wear. In turn, pyrolytic carbon has a high atrombogenicity. Unfortunately, ceramics are fragile and have low mechanical features, polymers are prone to degradation, and metals have low bioactivity coupled with the possible release of toxic substances into the human body. Carbonaceous materials are eroded in body fluids. Hence, one concept for improving the functionality of individual biomaterials is to combine them with others in the form of composites under the assumption that composite materials should have all of the positive features of their structures and at the same time attempting to reduce their negative features partially.

According to this assumption, one of the unintuitive but perspective bio(nano)composites can be silica functionalized by silver nanoparticles, which gradually release small amounts of metal ions from the matrix in long-term usage of such nanocomposite. Silica and its compounds are usually non-toxic, inert and, according to many literature studies, are materials that may improve the adhesion of other ceramics to different substrates, reduce long-term wear, prevent corrosion, limit the dissolution effects, and/or increase the stability of the ionic forms of metals [26,27,28]. In turn, silver-silica systems can be considered as antimicrobial agents or materials that improve both biocompatibility and bioactivity by facilitating calcium phosphates’ growth from bodily fluids. They may also create the possibility of a faster osteoconductive and osteoinductive action and increase the implantology potential of such synthesized structures in the future. Unfortunately, this kind of solution has a few problems that have to be considered during its further application. One of the most important factors is the toxicity effect of silver (i.e., 0.5 to 5 ppm for ions and 12.5 to 50 ppm for nanoparticles), limiting or even cause the death of human cells. In turn, silica forces thermal treatment to increase its adhesive potential, wherein this approach induces some structural modifications that may alter the physicochemical features of a newly formed biocomposite. When all of the pros and cons of inorganic bio(nano)materials that are prepared in this manner are considered, such materials can be a prospective new class of coatings. This is because there are still no solutions that solve the problem of the functionalization of a NiTi alloy using silica, silica-based, or silver-silica coatings.

The surface can be modified by many techniques such as deep-coating, spin-coating, sol–gel, plasma spraying, chemical vapor deposition, physical vapor deposition, etc. Each of these is based on different approaches to deposition, is specific for different materials, and enables coatings with different thicknesses or densities depending on the co-deposited material to be developed [29]. Unfortunately, there is still no one ideal method for modifying the surface, providing the multifunctionality of the coating and eliminating the necessity of the high-temperature processes. Some solutions that have been developed recently include the electrophoretic deposition technique (EPD), which is used to form simple and complex hybrid composite layers on substrates with irregular shapes, and/or morphologies [30,31,32,33,34]. EPD is an electrochemical method that makes it possible to control the thickness of the coating and deposit practically any biologically acceptable material, including organic polymers [20], carbides/nitrides, ceramics [19,35], and composites [36,37]. Unfortunately, EPD-formed ceramic coatings have a low level of adhesion (<10 MPa) to metallic substrates, which means that a post-heat treatment is required [38,39]. In turn, EPD is relatively cheap and does not cause damage to the structure of deposited materials. The high-temperature sintering of electrophoretically deposited ceramic coatings also has advantages, such as forming new biocompatible materials with unique properties [40,41]. Hence, new materials and functionalization approaches have to be considered for improving the quality of an implant.

Hence, the paper focuses on the functionalization of a medical NiTi alloy using a potentially new class of biologically active silver-silica composite. This approach is especially important due to the advantages and disadvantages of commonly used individual materials applied to modify the surface of biomaterials, recently. An interesting aspect discussed in the publication will also be the analysis of the impact of the temperature on the features of the coat-forming material, especially on its microstructure, chemical composition, and structural alterations. Another problem that is resolved in the paper is analyzing the structural modification on the border of the substrate and coating, and a description of the potential mechanism that leads to a structural reorganization. Thus, X-ray diffraction (XRD), Raman spectroscopy, X-ray photoelectron spectroscopy (XPS), and scanning electron microscopy (SEM) equipped with an energy dispersive spectrometer (EDS) were used to create a detailed characterization of the structure and morphology of a silver-silica coating as well as to determine the influence of sintering temperature on the mechanism of layer formation. These studies will assist in the development of an entirely new class of surface-functionalized metallic implants using inorganic coatings.

## 2. Experimental Part

### 2.1. Substrate Treatment Procedure

A commercially available NiTi alloy (Johnson Mathey, London, UK) in the β-phase (B2) with the characteristic temperatures of martensitic transformations below an ambient temperature was used as the substrate for the deposition of the hybrid coatings. The samples were polished and then passivated in a steam autoclave in an air atmosphere of 134 °C for 30 min to form a thin amorphous TiO_2_ layer [35]. The temperature was stabilized with an accuracy of 0.1 °C.

### 2.2. Suspension Preparation and Formation of the Coatings

The coatings were deposited from a colloidal suspension with a concentration of 0.1 wt.% SiO_2_/Ag nanocomposite powder in 50% ethanol (Avantor Performance Materials, Gliwice, Poland) using the electrophoresis (EPD) technique. The silver-silica nanosystem was prepared according to the procedure described by Peszke et al. [42]. Before deposition, the suspensions were placed into an ultrasonic bath for 2 h. The cataphoretic deposition was performed under a voltage of 5 V and durations of 1 to 15 min. Platinum was used as the counter electrode. Next, the coatings were dried at room temperature for 24 h. The uniform layers were then subjected to sintering at 700 °C in a technical argon atmosphere for 2 h to sinter the ceramic particles and increase the coating’s adhesion to the metallic NiTi substrate.

### 2.3. Coating Characterization

#### 2.3.1. SEM-EDS

TESCAN Mira 3 LMU scanning electron microscopy (SEM) equipped with an energy dispersive spectrometer (EDS) (TESCAN, Brno, Czech Republic) was used to determine the microstructure and chemical analysis of the obtained coatings. The chemical composition was estimated based on five points on the entire sample area, and the standard deviation was calculated based on this data. The images were collected by secondary electrons (SE). The samples, which were covered by a 10 nm chromium layer, were measured using a Quorum Q150T ES sputter coater (Quorum Technologies, East Sussex, UK).

#### 2.3.2. XPS Measurements

A Kratos AXIS Ultra DLD (Kratos Analytical Ltd., Manchester, UK) X-ray photoelectron spectrometer (XPS) was used to determine the electronic structure and chemical surface composition. The photoelectron spectra were collected using a monochromatized Al K_α_ radiation (1486.6 eV) from a depth of several nm. The anode’s power was set at 150 W, and the hemispherical electron energy analyzer was operated at a pass energy of 20 eV for all of the high-resolution elemental spectra. All of the measurements were performed with a charge neutralizer. The spectra were calibrated to the position of the aliphatic carbon C *1s* line position (Binding Energy (BE) = 284.8 eV). The data were analyzed using Kratos Vision (version 2, Kratos Analytical Ltd., Manchester, UK) software. The chemical composition was calculated from the area of the core lines, while the data were estimated based on the five points on the entire area of the sample. The standard deviation was calculated based on this data.

#### 2.3.3. XRD

An X’PertPro MPD PANalytical X-ray diffractometer (Malvern PANalytical, Almelo, The Netherlands) was used for the structural analysis. The coated NiTi alloy was examined by the grazing incidence X-ray diffraction technique (GIXRD). The GIXRD patterns were measured using Cu *K_α_* radiation at a constant incidence angle of 1.2° at room temperature. The qualitative analysis was performed using the HighScore Plus (version 4.8, Malvern PANalytical, Almelo, The Netherlands) software and the ICDD PDF 4+ database.

#### 2.3.4. Raman Measurements

A WITec confocal Raman microscope (CRM) alpha 300R (WITec Wissenschaftliche Instrumente und Technologie GmbH, Ulm, Germany) equipped with an air-cooled solid-state laser (λ = 532 nm) was used to perform the structural analysis as well as to determine the distribution of the material around the Ag-SiO_2_ coatings. The excitation laser radiation was coupled into a microscope via a 50 μm diameter single-mode optical fiber. The 50×/0.76 NA air Olympus MPLAN objective with a pinhole equal to 50 μm was used to preserve the lateral and depth resolutions. The Raman scattered light was focused onto a multi-mode fiber (50 μm diameter) and a monochromator with a 600 line/mm grating. The instrument calibration was verified by checking the position of the Si (520.7 cm^−1^). The lateral resolution was estimated according to the Rayleigh criterion LR = 0.61λ/NA, where LR is the minimum distance between the resolvable points (in the X-, Y-directions), NA is the numerical aperture, and λ is the wavelength of the laser excitation, which was equal to LR = 0.43 μm. All of the spectra were collected in the 200–4000 cm^−1^ range at 20 mW on a sample with a 3 cm^−1^ spectral resolution. The surface Raman imaging maps were collected in a (40 × 40) μm area using (160 × 160) pixels (=25,600 spectra) with an integration time of 30 ms per spectrum and precision of moving the sample ± 0.5 μm. The total exposure time for the Raman map was estimated ca. 40 min. The output data was manipulated by performing a baseline correction using the autopolynomial function of degree 3 and removing the cosmic rays. The chemical images were generated using a sum filter that integrated the intensity over a defined frequency range. In turn, the cluster analysis (CA) was used to group the individual objects (spectra from the map) into clusters. For this purpose, a K-means analysis with the Manhattan distance for all of the Raman imaging maps was performed. The Raman spectra obtained during the K-means cluster analysis were subjected to a band fitting analysis using the Voigt function in the GRAMS (version 9.2, Thermo Fisher Scientific, Waltham, MA, USA) software package. According to this analysis, one spectrum was found for the pristine sample, and three slightly different spectra (c1, c2, c3) were found for the sintered material. All of the analyses were performed on the normalized spectra using WITec Project FourPlus (version 4.1, WITec Wissenschaftliche Instrumente und Technologie GmbH, Ulm, Germany) software.

#### 2.3.5. Calorimetry

Differential scanning calorimetry (DSC) was used to study the influence of the deposition process on the course of the martensitic transformation. The measurements were taken using a DSC1 Mettler Toledo calorimeter (Mettler Toledo, Schwerzenbach, Switzerland). The heating/cooling rate of 10 °C per minute and an accuracy ±1 °C.

## 3. Results and Discussion

Microscopic observations were used to reveal the influence of the applied deposition parameters on the coat-forming material’s morphology and quality. The coatings that had been fabricated at a voltage of 5 V and short times (below 15 min) were heterogeneous, but the coating material did not cover the entire surface of the NiTi substrate. It was completely covered only after extending the deposition time to 15 min. The microstructure images illustrated large irregular agglomerates that were composed of submicron and nanometer irregular particles that were heterogeneously distributed around the coating. The area between its being observable at the nanometer scale was built by the homogenously distributed submicron- and nano- spherical-shaped particles (Figure 1a,b). The coating that was prepared in this manner had no discontinuities or cracks. The temperature effect usually strongly modifies the surface morphology and leads to a material consolidation that is represented by the coalescence of particles, which was also observed in this case. Looking more precisely at the nanoscale image, especially in the area between large agglomerates, some of the particles tended to agglomerate and form a continuous layer (Figure 1c,d). The chemical mapping illustrated that silicon and oxygen usually co-existed together and typically concentrated around the large agglomerates for both the pristine and sintered material. In turn, at the microscale, the carbon and silver were distributed homogenously without the tendency to agglomerate, while at the nanoscale submicron- and nano-spherical-shaped particles were primarily an effect of the co-existence of silver and oxygen with only an insignificant concentration of silicon (Figure 1c,f,g). A more detailed analysis that considered the nature of the individual elements will be discussed further in the paper. The titanium and nickel signal corresponded with the NiTi substrate and indicated that the deposited layer was relatively thin. After sintering, the signal from Ni became weaker, while Ti was stronger, which suggests an increase in the titanium oxide layer as a result of the influence of the temperature. Moreover, some of the silver particles had agglomerated after sintering (Figure 1e,g). The thermal-treated samples usually had a lower carbon content due to the decomposition of the organic matter, an increase in oxygen, or a decrease of the silver concentration due to evaporation [43]. Those data could indicate a chemical reorganization within the coating and the formation of a new composite layer.

An XPS analysis was performed to look more precisely at the chemical environment of the Ag-SiO_2_ surface and to analyze the impact of temperature on the outermost part of the co-forming material. The survey spectrum of the coat-forming material revealed the lines that are typically assigned to the silver-silica composite and the photoelectron lines, which had originated from the titanium and carbon. The quality of the co-deposited material was also confirmed by analyzing the atomic concentrations of the individual elements (Table 1). An interesting observation was that the Ag/O ratio slightly differed from the typical values for the silver oxides (1.00 or 0.50). One explanation might be the occurrence of silver in different chemical states or the atypical configuration of the silver at the surface, including the partially oxidized or metallic silver. In turn, the lower values of Si/O or Ti/O ratio relative to the theoretical ones for SiO_2_ and TiO_2_ (0.50) might correspond to the non-stoichiometric silica oxide (SiO_2−x_) as well as to the defected structure of titanium dioxide. However, a more detailed analysis of the individual photoelectron peaks is crucial to understanding the nature of the Ag-SiO_2_ coating’s surface.

The asymmetry in the Ag *3d* photoelectron line’s shape could indicate two possible components and different chemical states of silver. According to this assumption, two well-separated doublets with splitting values of about 6 eV were found (Figure 2). The doublet with the higher binding energies (Ag_5/2_ = 368.9 eV and Ag_3/2_ = 374.9 eV) may correspond to silver-organic complexes, and the second, much more intense doublet (Ag_5/2_ = 368.1 eV and Ag_3/2_ = 374.1 eV) to the metallic or ionic silver [44]. Unfortunately, the similar binding energy values of Ag^0^ and Ag^+^ made it difficult for their unambiguous assignment (Figure 2). Interestingly, some literature data has illustrated that silver in such environmental surroundings can occur in the form of sub-nanoparticles (<4 nm). A more detailed analysis will be discussed later in the paper.

Similar to the shape of the Ag *3d* photoelectron line, the strong asymmetry that was observed in the Si *2p* core line ensured fitting by the line at a binding energy of 101.9 eV, which corresponds to non-stoichiometric silica [45,46] or organosilicon compounds, e.g., the siloxy groups [47]. This interpretation was confirmed by analyzing the O *1s* core line within which three components that were located at 530.1 eV, 531.8 eV, and 533.3 eV were fitted. The component at the lowest binding energy was assigned to the lattice O bound to Ti^4+^ [48,49], whereas the other two peaks may be associated with the oxygen ions that are dispersed in an oxygen-deficient silica network, the carbonyl oxygen in the ester groups [50] or the oxide layer around the metallic silver [51]. The titanium line’s presence, which resulted from a passive interlayer that was fabricated on the NiTi substrate or from the migration of the titanium ion into the outer part of the surface, was interesting. This effect may suggest the porous nature of the Ag-SiO_2_ coating. In more detail, due to spin-orbit coupling, the Ti *2p* core lines were fitted using a doublet with a splitting-value equal to 5.6 eV, whereas the position of the maximum binding energy for Ti *2p*_3/2_ = 458.8 eV confirm the titanium dioxide. Finally, the C *1s* was deconvoluted according to Bik et al., and the carbon impurities that had been deposited on the surface of the Ag-SiO_2_ layer such as Si-C (283.7 eV), C-C/C-H (284.8 eV), Si-O-C (285.4 eV), C-O (286.5 eV), C=O (287.8 eV), and the ester groups (288.9 eV) or the remains of the synthesis were considered [52]. The presence of all of the individual elements was also observed during the SEM-EDS analysis (Figure 1c). It is worth noting that the XPS technique revealed that the surface had been enriched with carbon, silver, and silicon, while the lack of a nickel signal confirms the continuous character of the coating that had been postulated during the SEM analysis, and the same confirms the preferential oxidation of the titanium. Importantly, it is worth noting that the high carbon content makes interpreting the other lines problematic.

There were other interesting findings for the sintered sample. Similar to the previous observation, the survey spectrum highlighted that the main lines originated only from silver, silica, oxygen, titanium, and carbon. However, the atomic concentration of silver was estimated as two-fold lower than in the pristine material. The low atomic concentration of carbon, which was estimated based on the SEM-EDS, may correspond to the decomposition of the siloxy groups and silver-organic complex. In turn, a decrease in the silver content at the surface relative to the bulk may indicate that the silver had evaporated from the coating or had migrated into the deeper part of the material due to its high level of diffusivity. In turn, the higher atomic concentrations that were found for the oxygen and titanium may have resulted from surface oxidation. The atomic concentrations are summarized in Table 1.

Additionally, the Ag/O, Si/O, and Ti/O ratios differed from the values typical that are for the stoichiometric oxides, which may have resulted from the atypical configuration of the individual elements at the surface. According to this assumption, we might expect the occurrence of a synergistic effect of the defected silver oxides and metallic silver on the formation of the highly disordered silica network and titanium dioxide. However, a more detailed analysis of the individual photoelectron peaks is needed to understand the chemical alterations on the surface of the Ag-SiO_2_ coating due to the temperature.

Hence, similar to the pristine sample, the Ag *3d* core line was fitted using two well-separated doublets that corresponded to the silver oxides (367.3 eV) and the metallic or ionic silver (368.1 eV). Unfortunately, as was mentioned earlier, it is difficult to differentiate Ag^0^ from Ag^+^ (Figure 2). There was also some modification of the pristine material for the Si *2p* and O *1s* core lines. More precisely, two components resulting from the non-stoichiometric silica (102.1 eV) and the fully oxidized porous silicon layer (103.7 eV) were found due to the fitting of the Si *2p* core line. In turn, the deconvolution of the O *1s* core line revealed three components with their maxima centered at 530.1 eV, 531.8 eV, and 533.3 eV. The assignment of the lowest binding energy component is problematic due to the similar binding energy characteristics for Ag(I)O/Ag(II)O and/or TiO_2_, whereas the other two lines might be assigned to the adsorbed hydroxyl species (OH^−^) on the titanium dioxide surface [53] or may correspond to the Si-O-Ti [54,55], respectively. However, considering that the samples were sintered at a high temperature, the probability of the adsorption of the OH^−^ groups was minimal and might have been associated only with the surface. In turn, the interpretation of the presence of the Si-O-Ti linkage seems to be burdened with a high degree of uncertainty, and another technique was required to confirm this hypothesis. The different chemical environment that was observed for the sintered material was also reflected by the interpretation of the Ti *2p* core line. Here, the most intensive line (459.2 eV) was assigned to the titanium dioxide, while an atypical line at higher binding energy (460.3 eV) may correlate with the surface modification within the TiO_2_ interlayer or the formation of the structurally defected titanium dioxide [56]. The stronger titanium signal of the deposited coating could have resulted from the microporosity that formed due to the surface infiltration of water vapor; however, the continuity of the layer was not broken according to the SEM observations (Figure 1d). This effect can be explained by a substantial reorganization on the border of the silver-silica and amorphous titanium layers. The analysis of the carbon C *1s* peak regarding the thermally treated sample should be considered in the context of possible carbon impurities such as Si-C (283.6 eV), hydrocarbon (288.8 eV), Si-O-C (285.4 eV), C-O (286.5 eV), C=O (287.8 eV), and COO-R (288.9 eV).

The chemical and surface analyses were supplemented by a structural analysis using XRD and Raman spectroscopy approaches. As a result, representative XRD pattern revealed the crystalline phase B2 of the NiTi alloy with a cubic symmetry (Fd-3m) and an amorphous and/or nanocrystalline phase of the silver-silica nanocomposite, which was visible as a distinct increase in the background in the range of 14–27° 2θ (Figure 3). There were no diffraction lines for the other phases. Similar diffraction patterns were found for the coatings after sintering (Figure 3) except that a Ti_2_Ni with a cubic lattice (Fd-3m) appeared due to the partial decomposition of the NiTi alloy. Similar observations were previously reported in the literature [12,57,58]. There was no visible evidence of metallic silver or other silver phases, which suggests that their nanometer size or the presence of silver was below the diffraction detection limit (Figure 3). Therefore, to look more precisely at the structural features of the coat-forming material before and after sintering, Raman spectroscopy was used.

The Raman spectrum of the pristine material was characterized by many more or less intense bands, wherein according to the literature reports, most of them (285, 714, 1064, 1372, 1510 cm^−1^) originated from the stretching and deformation modes within the (CO_3_)^2−^ units in the structure of the silver carbonate (Figure 3, upper right panel) [59,60]. The interpretation of the silver carbonate origin at first glance is, however, difficult to explain. One hypothesis could refer to the formation of Ag_2_CO_3_ as early as at the synthesis stage of the silver-silica nanocomposite, wherein the silver carbonate can co-exist with metallic silver (Ag^0^) [42], silver oxides (Ag(I)O, Ag(II)O) [61] and/or the silver-silicates system (Ag_6_Si_2_O_7_) [62]. In this stage, we have to assume that the exchange reaction occurred in the colloidal suspension between the silver nitrate and the mixture of sodium hydroxide with sodium carbonate. Another hypothesis could be related to the so-called carbonation in the water solution and can be considered analogously to the C-S-H system [63,64]. According to this assumption, during the electrophoretic deposition, the silver-silica nanosystem might show chemical activity and strongly react with the air’s carbon dioxide or might have originated from the partial oxidation of ethanol into the carbon dioxide during the cataphoretic deposition process. When this occurs, some fraction of unbounded silver oxide might undergo carbonation, leaving the initial silver-silica system unchanged. Here, the colloidal suspension before the electrophoretic deposition was stimulated by ultrasounds that caused an increase in the temperature of the reaction environment. A similar effect could occur during the EPD process as a result of the applied voltage. Confirmation of these hypotheses might also be the co-existence of the silver carbonate bands with another band derived from the silver oxides and silica network located at 222, 403, 466, 898, 933, 982, 1025 cm^−1^ [61,65]. In this context, the strong silica bands located above 800 cm^−1^ refer to the silicon-oxygen tetrahedral *Q^n^* units (n = 0–4 and stand for the amount of bridging oxygen per SiO_4_ tetrahedron). They thus indicate the depolymerization of the silica network [66]. An interesting observation related to the Raman spectrum of the deposited material is the presence of the atypical bands that were located around 2110 cm^−1^ with a shoulder at 2172 cm^−1^ and two bands around 2862 and 2923 cm^−1^. The first two bands’ nature is very difficult to explain unambiguously, while the second pair corresponded to the symmetric and asymmetric stretching vibrations of the CH_x_ (x = 2,3) groups. One hypothesis for explaining the origin of the low-lying bands is given in the literature reports, which suggests the adsorption of the CO molecules on the low-coordinated gold nanoparticles (Au^0^ or Au^δ+^ sites in the defect positions) [67,68] or nanocrystalline copper [69]. Analogous to these results, the appearance of a band at 2110 cm^−1^ that had a higher intensity might suggest CO’s adsorption on the surface of the metallic silver nanoparticles in the Ag defect position. In turn, the low intense band at a higher wavenumber might correspond to the gathering of a positive charge at the Ag^δ+^ surface. This hypothesis may prove the previous assumption, which was associated with a highly reactive reaction environment. Still, it may also enforce the presence of nano-dimensional metallic silver nanoparticles in the solution and within the coat-forming material. This observation also confirms the absence of any signal during the XRD analysis (Figure 3). The origin of the metallic silver may be explained as being two-fold: (i) considering the reaction conditions during the fabrication of the initial silver-silica composite; and/or (ii) the decomposition of ethanol from the suspension during the EPD process and the reaction with Ag(I)O that originated from the silver-silica composite. The correctness of both assumptions was chemically supported and confirmed by the XPS studies conducted on the coat-forming material (Figure 2). Finally, the low intense bands resulted from the ν(CH_x_) groups could be explained by assuming the formation of the silver-organic complex or low molecular weight siloxy units during the electrophoretic deposition. According to the first assumption, the silver-organic complex might have resulted from the highly reactive environmental conditions as was mentioned earlier, while the siloxy units might have been formed as a result of the reaction of the acetic aldehyde that was left after the ethanol reaction with silver oxide. Here, the functional units of the acetic aldehyde probably reacted with the functional groups of the strongly disordered silica network.

A very complex analysis of the Raman spectra, especially considering the silver within the deposited coating and the sintered one, implies the problem of the potential heterogeneity of the distribution of the individual phases in the coat-forming material. Because of this assumption, a more precise analysis using Raman imaging was performed. An integrated intensity analysis was conducted in relation to the bands of four regions: (1) 215–255 cm^−1^, (2) 445–495 cm^−1^, (3), 595–635 cm^−1^, and (4) 1045–1095 cm^−1^. Due to their high specificity, all of those bands should provide information about the distribution of the phases on the entire area of the coat-forming material (Figure 4). Moreover, the imaging analysis is presented on an exemplary set of data, whereby the information statistically was the same when considering any fragment of the alloy.

The Raman imaging maps that were created based on an integrated intensity analysis of the bands (1), (2), (4) for the pristine Ag-SiO_2_ sample had strong and medium-strong yellow color spots, which were associated with the silver oxide, silica, and silver-carbonate, respectively. All of these phases were dispersed homogeneously on the entire area, while in some places, they formed agglomerated, a few micrometer-sized objects. The fact that the strongest yellow spot signals of the Ag-O and Ag_2_CO_3_ did not always correspond to each other, indicates a multi-phase spatial structure of the material (Figure 4). In turn, the integrated intensity analysis of the bands from the region (3) returned almost dark Raman images with a signal close to the noise level, thus illustrating a lack of a structural alteration on the border of the Ag-SiO_2_ coating and TiO_2_ interlayer.

A different structural image was found regarding the sintered material (Figure 3). Here, the Raman spectra were grouped into three slightly different clusters (c1, c2, c3), while the main differences between them were related to the intensity of the variable bands, suggesting the occurrence of the structural heterogeneity within the coat-forming material. Other interesting findings were the complicated and irregular shape of the Raman spectra. According to the fitting analysis, many individual bands with a high full width at half maximum (FWHM) indicated that an atypical molecular structure had been formed due to the sintering. A similar Raman band arrangement was previously reported for the sintered hybrid Ag-SiO_2_/β-TCP system, which had been deposited on the TiO_2_/NiTi surface [40].

The spectra of sintered Ag-SiO_2_ coating may also be characterized by four bands whose nature is relatively easy to explain and many others whose origin is difficult to interpret. According to this assumption, the two bands centered at 240 cm^−1^ and 450 cm^−1^ might have originated from the Ag-O and Si-O vibrations, respectively, while the other two bands that were located around 395 and 615 cm^−1^ could indicate the crystallization of the rutile (Figure 3). The intensity of the Ag-O bond was the same before and after sintering, while the intensity of the silica band increased relative to the initial material. These findings might correspond to the similar bulk content of silver oxide in the analyzed area and its rearrangement within the silica network due to the sintering. In turn, the appearance of the rutile bands could be associated with the occurrence of micropores in the Ag-SiO_2_ coating; the low coating thickness, which was much below the depth resolution of the Raman spectrometer (~900 nm) or reorganization of the material on the border of the silver-silica coating and the amorphous titanium interlayer. The other bands were probably an effect of the silver nanoparticles’ interaction on the amorphous silica and titanium dioxide, which formed an atypical molecular structure composed of the combination SiO_2_-TiO_2_ [40]. In this context, silver acting as a catalyst probably stimulated the structural reorganization of the amorphous phases by lowering the temperature required to form such an atypical composite. As a result of the structural reorganization and high mobility of silver, we can also hypothesize that the silver ions could migrate into the newly formed amorphous interlayer and occupy the interstitial position within the silica-titanium network. This hypothesis could be confirmed using an XPS analysis with information about the presence of titanium dioxide and the Si-O-Ti bonds (Figure 1). Finally, the absence of bands of the CO that had adsorbed on the surface of the silver nanoparticles or siloxy units was an effect of its decomposition due to the temperature (Figure 3). Organic structure decomposition might provide the appearance of the presence of the low intense bands between 1300–1500 cm^−^^1^. Alternatively, these bands might have originated because of the incorporation of the amorphous carbon into the structure of coat-forming material due to the temperature and its structural rearrangement within silica and titanium dioxide.

The high complexity of the Raman spectra and the presence of three clusters could indicate a phase diversity in the area of the coat-forming material. Here, the integrated intensity analysis of the bands from regions (1) and (2) revealed a strong signal of the silver oxide and silica within the studied area, which suggests a lack of structural decomposition of the Ag-SiO_2_ (Figure 4). However, a more precise analysis of the band’s intensity for a sample before and after annealing revealed some interesting trends. A similar intense signal of the silica in the considered area revealed its homogeneous distribution in the studied area, while the slight increase in the signal associated with the deposited material indicates a low impact of temperature on its spatial arrangement (e.g., agglomeration). On the other hand, an almost two-fold increase in the silver oxide’s signal intensity for the pristine material and the presence of irregular strong, yellow-colored spots indicated a heterogeneous distribution of the silver oxides in the studied area. Unfortunately, the increase in the Ag-O signal was difficult to explain, and two alternative hypotheses had to be considered. One of them was associated with the surface oxidation due to temperature, while another one corresponded to the thermal decomposition of the silver carbonate and the formation of Ag(I)O [70]. Those data correlate with previous observations obtained from the XPS analysis quite well, thereby illustrating the formation of the silver oxides and the oxidized silica layer (Figure 2). Other interesting findings were provided by analyzing the imaging maps that were created based on the integrated intensity of bands (3) 595–635 cm^−^^1^ and (4) 1045–1095 cm^−^^1^. Here, the yellow-brown spots that were heterogeneously distributed on the entire area indicated the formation of titanium dioxide, while the completely dark image suggested a silver carbonate-free area (Figure 4). The high correlation of the titanium dioxide signal with the silica signal is interesting and might confirm a structural rearrangement within the entire studied area that had occurred on the border of the Ag-SiO_2_ coating and the TiO_2_ interlayer due to the temperature.

Finally, the formation of an atypical amorphous interlayer did not affect the blocking of the martensitic transformation and the shape memory effect (Figure 5). Hence, the transformation between the B2 parent phase and the monoclinic martensite B19′ is a two-step process that occurs through the R-phase during the cooling, i.e., the forward transformation B2↔R is separated from R↔B19′. The DSC studies also revealed the occurrence of a one-step reversible martensitic transformation that began at B19′ and ended at B2. The characteristic transformation temperatures that were determined are presented in Figure 5 and Table 2. Similar data were previously found for HAp and BCP coatings that functionalized the NiTi alloy [12,58]. Those results indicate the possible importance of Ag-SiO_2_ as a new class of coat-forming materials.

## 4. Conclusions

The electrophoretic deposition was used to functionalize a passivated NiTi alloy surface using a chemically fabricated Ag-SiO_2_ nanocomposite. A crack-free coating that covered the entire surface of the alloy was formed at 5 V for 15 min. SEM-EDS studies revealed that the coat-forming material in a micrometer scale is agglomerated in the larger or smaller irregular objects consisted of silica and a film of submicron and nano- spherical-shaped particles, which consisted of silver, carbon, and oxide. According to the XPS, the surface was enriched with carbon (62.5%) and silver (15.1%) and a relatively low concentration of silicon (3.8%) and titanium (1.2%). In turn, the more volumetric SEM-EDS data showed an elevated concentration of oxygen (22.6%) and titanium (31.3%) that had originated from the passivated titanium dioxide layer and low content of silver (1.9%) and silicon (1.9%) of the coating. These data clearly indicated the relatively thin coat-forming materials, that structurally were composed of silver carbonate, silver oxide, and silica structures. A sintering temperature of 700 °C per 2 h in a protective argon atmosphere was used to consolidate the substrate’s coat-forming material. According to the procedure, there was a coalescence of the ceramic particles, a chemical modification, and a structural reorganization of the deposited amorphous silver-silica nanocomposite observed. The interesting observations were the surface oxidation (38.5%), the increase of the titanium content (8.1%), and the decrease of the silver content (8.4%). Similar observations were made after a more volumetric data analysis with a high content of oxygen (64.2%), titanium (27.6%), and low concentrations of silver (2.4%) and silica (0.9%). All of this data indicates the oxidation of the titanium dioxide interlayer and the formation of a porous structure within the coat-forming materials. Additionally, the surface, which was enriched with carbon (41.9%) relative to the bulk (1.4%), suggests the incorporation of carbon into the silver-silica coating. The XPS and Raman investigations indicate the formation of a new kind of an atypical glass-like phase that consisted of SiO_2_-TiO_2_ with silver ions that were incorporated into the interstitial position of the network. This kind of new coating did not block the martensitic transformation, which is responsible for the shape memory effect. Hence, the two-step martensitic transformation during the cooling first occurs in the form of B2↔R and then R↔B19′ while the reversible transformation is in the form of B19′ to B2.

## Figures and Tables

**Figure 1 materials-14-00099-f001:**
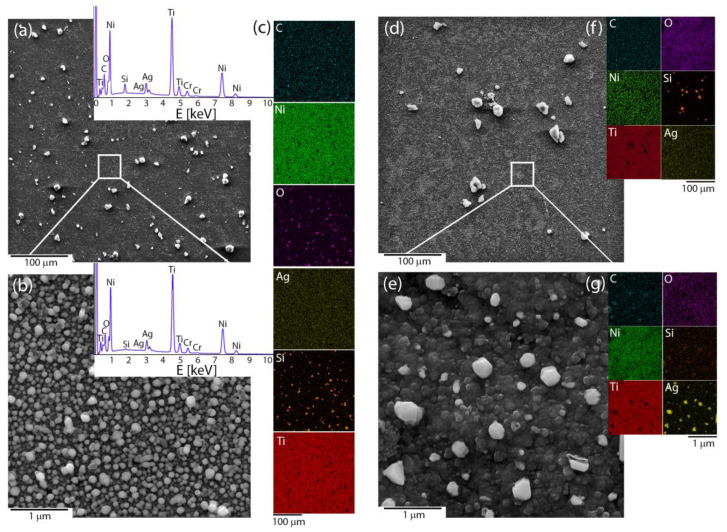
Microstructures of the (**a**,**b**) pristine and (**d**,**e**) sintered Ag-SiO_2_ coating that was obtained based on the SEM images at two different scales, (**c**,**f**,**g**) chemical composition imaging of an individual elements. The EDS spectra from selected areas had a similar character for the pristine and sintered material.

**Figure 2 materials-14-00099-f002:**
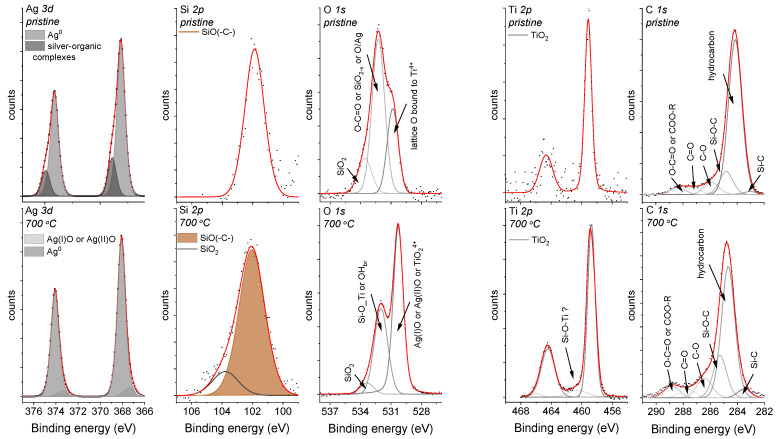
The Ag *3d*, Si *2p*, O *1s*, Ti *2p,* and C *1s* core levels for the Ag-SiO_2_ coatings that had been deposited on the TiO_2_/NiTi substrate that was obtained before (**upper panel**) and after (**lower panel**) sintering. The core levels were fitted using the Voigt function, and the background for each peak was subtracted using the Shirley baseline.

**Figure 3 materials-14-00099-f003:**
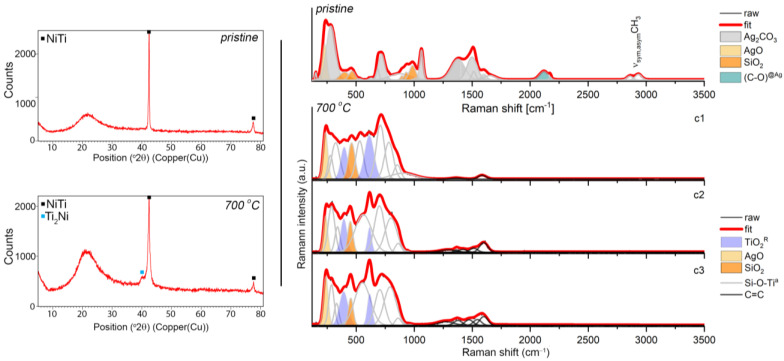
XRD data (**left panels**) and Raman spectra (**right panels**) that were extracted from the *K*-means cluster analysis considering the pristine Ag-SiO_2_ composite and the composite that was sintered at 700 °C per 2 h deposited on TiO_2_/NiTi substrate. The band fitting of the Raman data was analyzed using the Grams 9.2 software package and the Voigt function, and the individual chemical phases are color-marked.

**Figure 4 materials-14-00099-f004:**
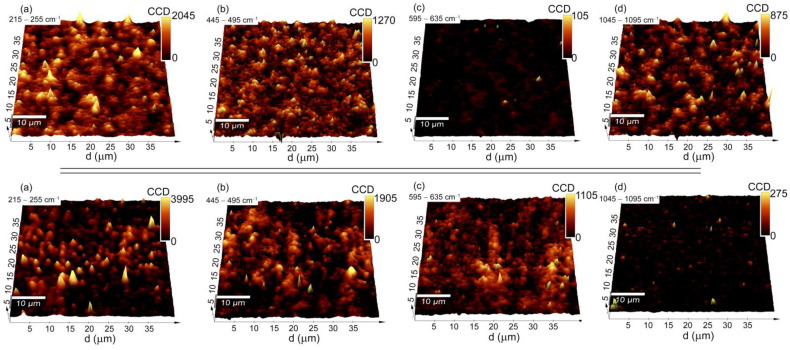
3D Raman imaging maps of the pristine Ag-SiO_2_/TiO_2_/NiTi composite (**upper panels**) and the composite that was sintered at 700 °C (**lower panels**) that were obtained as a result of the integrated intensity analysis considering the bands that were assigned to (**a**) Ag-O; (**b**) SiO_2_; (**c**) Si-O-Ti; and (**d**) Ag_2_CO_3_.

**Figure 5 materials-14-00099-f005:**
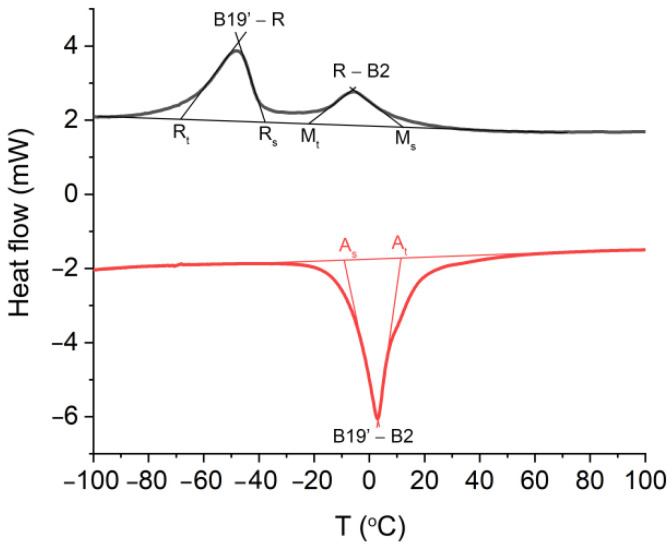
Cooling (black)/heating (red) curves measured for the NiTi alloy that was sintered at 700 °C per 2 h. A_s_—start temperature of the reverse martensitic transformation, A_f_—finish temperature of the reverse martensitic transformation, M_s_-start temperature of the forward martensitic transformation, M_f_—finish temperature of the forward martensitic transformation, R_s_—start temperature of the B2→R transition, R_f_—finish temperature of the B2→R transition, B2→R→B19′—martensitic transformation, B19′→B2—reverse martensitic transformation.

**Table 1 materials-14-00099-t001:** Atomic percentage concentration of the individual elements of the coat-forming materials. Data originated from the modified NiTi alloy before and after sintering at 700 °C per 2 h. The SEM-EDS should be considered to be a semi-qualitative method in this analysis. The data in brackets refers to the standard deviation data, which was obtained based on the measurements of the five different points.

	Pristine		Sintered at 700 °C	
	XPS	SEM-EDS	XPS	SEM-EDS
	at.%	at.%	at.%	at.%
O	17.4 (1)	22.6 (4)	38.5	64.2
Ti	1.2 (1)	31.3	8.1	27.6
Ag	15.1 (1)	1.9	8.4	2.4
C	62.5 (1)	11.7	41.9	1.4
Si	3.8 (1)	1.9	3.1	0.9
Ni	–	30.6	–	3.5

**Table 2 materials-14-00099-t002:** Transformation temperatures of the modified NiTi alloy after sintering. A_s_—start temperature of the reverse martensitic transformation, A_f_—finish temperature of the reverse martensitic transformation, M_s_—start temperature of the forward martensitic transformation, M_f_—finish temperature of the forward martensitic transformation, R_s_—start temperature of the B2→R transition, R_f_—finish temperature of the B2→R transition, B2→R→B19—martensitic transformation, B19→B2—reverse martensitic transformation.

A_s_ (°C)	A_f_ (°C)	R_s_ (°C)	R_f_ (°C)	M_s_ (°C)	M_f_ (°C)
−7.3	10.5	9.2	−17.2	−39.2	−65.0

## Data Availability

Data are stored at the cloud and stick in the form of backup.

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
