# Peer review of "Innovative Bioactive Ag-SiO2/TiO2 Coating on a NiTi Shape Memory Alloy: Structure and Mechanism of Its Formation"

_materials, 2020, doi:10.3390/ma14010099_

Round 1
Reviewer 1 Report
The paper introduces a new class of materials (Ag-SiO2) for coating NiTi alloys. Systematic experiments are taken, and chemical, surface and structural analyses are conducted to reveal the micro-structure and formation mechanism of the coating. The paper can be of interest to the readers of this journal. Grammar mistakes should be corrected, e.g. Line 41 – “inducing” should be “induced”, Line 58 – “crucial is to note” should be “it is crucial to note”. Moreover, figures should be renumbered since Fig. 3 is before Figs. 1 and 2.
Author Response
The paper introduces a new class of materials (Ag-SiO2) for coating NiTi alloys. Systematic experiments are taken, and chemical, surface and structural analyses are conducted to reveal the micro-structure and formation mechanism of the coating. The paper can be of interest to the readers of this journal. Grammar mistakes should be corrected, e.g. Line 41 – “inducing” should be “induced”, Line 58 – “crucial is to note” should be “it is crucial to note”. Moreover, figures should be renumbered since Fig. 3 is before Figs. 1 and 2.
I would like to thank you the Reviewer for the comment. All of the mistakes were corrected in the newest version.
Reviewer 2 Report
- The article still needs some grammatical and syntax improvements. Use of English service center is recommended.
- The introduction needs to be revised for higher quality language. Using separated paragraphs is encouraged but with more details. In addition, the authors mentioned some works without stating about the contributions, pros and cons and the how the current work would address.
- The purpose of this study and applications of it should be delineated clearly in Introduction
- The codes based on which the preparations are made should be mentioned.
- What is the reason or justification for the increase in the Ag-O signal?
- The authors mentioned about the “Metallic materials (e.g. steel, titanium, tantalum, niobium, precious metals) and their alloys have 37 been interested in medicine and especially in implantology for over 65 years”, for the new materials the following more recent work are recommend to be considered.
- Farzampour, A. (2019). Compressive behavior of concrete under environmental effects. IntechOpen.on Gene Expression Programming. International Journal of Steel Structures, 1-11.
- In conclusion section, more quantitative results related to the study should be added. It is recommend to avoid other studies results representation in conclusions
- Too many references from the same authors should be avoided. Instead other similar works should be considered.
Author Response
The article still needs some grammatical and syntax improvements. Use of English service center is recommended.
I would like to the valuable comment. The text in the latest version had been corrected by the native speaker.
The introduction needs to be revised for higher quality language. Using separated paragraphs is encouraged but with more details. In addition, the authors mentioned some works without stating about the contributions, pros and cons and the how the current work would address.
I would like to the valuable comments. The text in the latest version had been corrected according to the Reviewers’ suggestion.
The purpose of this study and applications of it should be delineated clearly in Introduction
It has been corrected according to the Reviewers’ suggestion.
The codes based on which the preparations are made should be mentioned.
It has been supplemented in the Experimental part.
What is the reason or justification for the increase in the Ag-O signal?
This effect had been discussed during the Raman analysis. You can find this information in lines 418-426 and also below:
From the other side, an almost twice increase in the signal intensity of silver oxide concerning the pristine material as well as the presence of irregular strong yellow-colored spots pointed out the heterogeneous distribution of silver oxides in the studied area. Unfortunately, the increase in the Ag-O signal was difficult to explain and two alternative hypotheses had to be taken into consideration. One of them refers to the surface oxidation due to temperature, while another one corresponds to the thermal decomposition of silver carbonate and the formation of silver(I) oxide [60]. Those data quite well correlate with previous observations obtained from XPS analysis illustrating the formation of silver oxides and oxidized silica layer (Fig. 2).
The authors mentioned about the “Metallic materials (e.g. steel, titanium, tantalum, niobium, precious metals) and their alloys have 37 been interested in medicine and especially in implantology for over 65 years”, for the new materials the following more recent work are recommend to be considered.
We would like to thank you for the comment. We would like to emphasize that the paper had been focused not on the development of new substrates but only on the formation of coatings that can improve the functionality of the commonly used materials. Therefore, the aspect of substrates as new potential implant materials had been discussed very shortly, while more in detail the advantages and disadvantages of commonly used NiTi alloy had been discussed. Of course, you have right the information about potential new implant biomaterials had been missing and some additional information has been added in the newest version.
Farzampour, A. (2019). Compressive behavior of concrete under environmental effects. IntechOpen.on Gene Expression Programming. International Journal of Steel Structures, 1-11.
In conclusion section, more quantitative results related to the study should be added. It is recommend to avoid other studies results representation in conclusions
Thank you for the comment. The conclusions had been improved according to the Reviewers’ suggestion.
Too many references from the same authors should be avoided. Instead other similar works should be considered.
Thank you for the comment. You have absolutely right, but unfortunately, it is difficult to avoid these references because our group is one of the teams working with the functionalization of the NiTi alloy by inorganic coatings around the world. Other teams are working with titanium alloys or other kinds of titanium-based alloys. Another problem is that during the substrates’ functionalization, we are basing on the silver-silica materials prepared by ourselves. Of course, we can cite other papers that taken into account the problem of functionalization of the surface of other biomaterials but it can be problematic in the data interpretation because of the difficulties related to the substrate and interaction between deposited coat-forming materials and the substrate. We can also cite other papers considered the preparation of silver-silica structures but in these papers inorganic nanocomposites were prepared in another way, using other reagents and laboratory conditions. In consequence, another type of structure with completely different sizes and shapes of silver nanoparticles or silver nanoparticles' degree of distribution around the matrix had been prepared. Additionally, we would like to emphasize that many coatings, will behave completely different from another kind of substrate, and crucial is to refer to the material which is the subject of the studies. Therefore, despite many autoreferences, as you mentioned, all of that papers are focused on the fabrication of coatings on NiTi alloy, while others are focused on the formation of silver-silica nanocomposite that is used for further functionalization of the substrate. All of those papers are strongly consistent with each other and there is part of the whole.
Reviewer 3 Report
The manuscript titled "Innovative bioactive AgSiO2/TiO2 coating on NiTi shape memory alloy: Structure and mechanism of formation " is a well written paper. There are a few issues, however that need to be addressed (please see the commented manuscript attached in the peer review). The idea and approach are highly interesting, but the presentation could be further improved. Statistical significance should be included in the paper. In my opinion, the manuscript deserves to be considered for publication.
//Accept after semi-Major Revision

Author Response
All modifications proposed by the Reviewer have been implemented into the newest version of the text, while our comments are in the pdf file.

Reviewer 4 Report
In the article “Innovative bioactive Ag-SiO2/TiO2 coating on NiTi shape memory alloy: Structure and mechanism of formation” detailed study of the complicated structure of the coatings was carried out. The main goal of the article was to reveal the structure of the coatings and the mechanism of their formation. However, neither task was completely solved because objective research results presented in the article are not enough.
- There is no information about the thickness of the coatings. Was it determined, by what method and what are the specific results?
- SEM-images (or TEM – images) of the cross-section of coating, as well as elements distribution maps in them are absent. This information could be key in identifying the mechanism of the coating formation.
- XRD analyses was found the structure of the coating was amorphous before the heat treatment and glass-like after that. The presence of nanocrystallites of different compounds is possible. The authors tried to reveal them when performed detailed research using methods XPS and Raman measurements. However, these methods give indirect results that are difficult for interpretation. This could be established quite accurately with help of the TEM method.
- Authors reported that phase composition of the coating included silver oxide and Ag2CO3 in amorphous SiO2 before the heating (Fig. 4 upper panels) and “glass-like phase consisted of SiO2-TiO2 with silver ions” (line 452) after that. However, both the XPS and SEM-EDS results showed a very low Si content. It is possible that Si is localized mainly in the particles observed on the surface of the coatings in SEM images.
- The elemental analysis results obtained by the XPS and EDS methods are dramatically contradictory. How much Ag is contained in the coatings exactly? This point is very important since the toxicological properties of the coatings strongly depend on the amount of silver.
- Figure number 3 is presented first of all in the article and not after Figure number 2. Why?
Author Response
- There is no information about the thickness of the coatings. Was it determined, by what method and what are the specific results?
Unfortunately, the thickness of the coatings has not been measured. Using the classic methods of making the cross-section (cutting + including + grinding and mechanical polishing), it was not possible to obtain good quality samples to estimate the thickness of the coatings. Based on the results from SEM + EDS, we can only hypothesize/conclude that the layer between the agglomerates is relatively thin. Both before sintering and after heat treatment, no Ti and Ni derived from the NiTi substrate were identified in the agglomerate sites.
From the tests carried out with the use of a contact profilometer for the sintered layer (Rz = 4.3±0.5 μm and Rt = 5.2±0.5 μm), we can conclude that the agglomerates had a height of max. approx. 5 μm. At the places of the scratched layer/layer, the roughness parameters related to the profile height were comparable with the parameters measured for the unmodified NiTi substrate (Rz = 0.25 ± 0.05 μm and Rt = 0.32 ± 0.05 μm), therefore on this basis, it is difficult to determine the thickness of the thin layer between the agglomerates.
- SEM-images (or TEM – images) of the cross-section of coating, as well as elements distribution maps in them are absent. This information could be key in identifying the mechanism of the coating formation.
We fully agree that such a study could provide interesting information, for example on the diffusion of individual elements across the layer. However, the measurements were not taken for the reasons mentioned above.
- XRD analyses was found the structure of the coating was amorphous before the heat treatment and glass-like after that. The presence of nanocrystallites of different compounds is possible. The authors tried to reveal them when performed detailed research using methods XPS and Raman measurements. However, these methods give indirect results that are difficult for interpretation. This could be established quite accurately with help of the TEM method.
We agree with the Reviewer. However, due to the inability to prepare a thin film for TEM investigations, such measurements were not made. The starting material used for electrophoretic deposition was characterized, among others, by using TEM and research has shown that it is nanocrystalline. The results were published [10.1039/C7RA00720E]. From the SEM images, we can also indirectly conclude that the phases visible on the surface of the layer are of submicron and nanometric size.
At the same time, we would like to emphasize full agreement with the Reviewer. TEM measurements would provide additional interesting structural information. Unfortunately, at the moment, we do not have access to specialized equipment. In research projects, we have applied for funding for this type of research.
- Authors reported that phase composition of the coating included silver oxide and Ag2CO3in amorphous SiO2 before the heating (Fig. 4 upper panels) and “glass-like phase consisted of SiO2-TiO2 with silver ions” (line 452) after that. However, both the XPS and SEM-EDS results showed a very low Si content. It is possible that Si is localized mainly in the particles observed on the surface of the coatings in SEM images.
Yes, it is. Moreover, silicon is mainly localized in the big agglomerates. Other phases detected by Raman or XPS analysis are an effect of the specificity of each technique.
- The elemental analysis results obtained by the XPS and EDS methods are dramatically contradictory. How much Ag is contained in the coatings exactly? This point is very important since the toxicological properties of the coatings strongly depend on the amount of silver.
The differences in the XPS and EDS methods result from the analysis of the material from different depths. XPS is a technique typical for surface examination (3-5nm). Moreover, in the EDS results, the table includes Ni from the substrate and Ti (substrate + passivated layer) (probably a few micrometers). After the EDS results normalization, the Ag values are closer to each other.
Of course, we agree that the Ag content strongly influences the biocompatibility of the coatings and their antibacterial properties. This is widely discussed in the publication which is now under review.
- Figure number 3 is presented first of all in the article and not after Figure number 2. Why?
It has been corrected.
Round 2
Reviewer 2 Report
The article still needs extensive grammatical and syntax improvements. Many terms and sentences should be changed to be formal and appropriate.
Many references from the same author or authors. These references should be changed for more recent refs.
For introducing the metallic dampers in introduction the following ref is recommended for consideration:
Farzampour, A., Mansouri, I., Mortazavi, S. J., & Hu, J. W. (2020). Force–Displacement Relationship of the Butterfly-Shaped Beams Based on Gene Expression Programming. International Journal of Steel Structures, 20(6), 2009-2019.
Author Response
The article still needs extensive grammatical and syntax improvements. Many terms and sentences should be changed to be formal and appropriate.
Dear Reviewer. We have done all the best to improve our English in the paper. The paper has been proofread by all of the co-authors, the Grammarly app, and finally by the English native speaker (all small changes have been yellow-marked). I don't know what we can do more?
Many references from the same author or authors. These references should be changed for more recent refs.
I know that we have self-cited many publications prepared previously by ourselves, but as we mentioned before all publications are focused on the NiTi shape memory alloys or silver-silica nanomaterials developed especially for NiTi alloy functionalization. All of these studies are part of the scientific project and therefore all studies are consistent to receive the new kind of bioactive biomaterial for further medical application. In our opinion, it is difficult to cite other papers, e.g. coatings on steel or titanium or titanium-based derivative substrates because of the substrate differences (structural and chemical). As a result, completely different behavior is expected to obtain if we cover the biomaterial by proposed coatings. Of course, if it is a huge problem for you, we can remove some of our references.
For introducing the metallic dampers in introduction the following ref is recommended for consideration:
Farzampour, A., Mansouri, I., Mortazavi, S. J., & Hu, J. W. (2020). Force–Displacement Relationship of the Butterfly-Shaped Beams Based on Gene Expression Programming. International Journal of Steel Structures, 20(6), 2009-2019.
I would like to thank you for the proposition but after a detailed inspection of the information of the paper, I cannot agree to cite them because of the different themes that are a subject of the paper in relation to our studies.

Reviewer 4 Report
The Authors tried to provide comprehensive answers to the comments.
Author Response
We would like to thank you the Reviewer for understanding and accepting our replies.
Round 3
Reviewer 2 Report
NA.